# Role of Senescence-Resumed Proliferation in Keloid Pathogenesis

**Ching-Yun Wang** [1], **Chieh-Wen Wu** [2] **and Ting-Yi Lin** [3,*]

1 School of Post-Baccalaureate Medicine, Kaohsiung Medical University, Kaohsiung 807378, Taiwan
2 Department of General Medicine, Kaohsiung Medical University Hospital, Kaohsiung Medical University, Kaohsiung 807378, Taiwan
3 Doctoral Degree Program of Translational Medicine, Taiwan Institute of Biomedical Sciences, National Yang Ming Chiao Tung University and Academia Sinica, Academia Sinica, Taipei 114514, Taiwan
* Correspondence: lintingyi2014@gmail.com

**Abstract:** Senescence-resumed proliferation (SRP) is proposed to be a mechanism associated with the escape of p21-mediated senescence and the activation of Wnt/β-catenin pathways that enhance malignancy. The keloid genomic landscape shows heavy intersections between TP53 and TGF-β signaling. The machinery to maintain cellular integrity through senescence, apoptosis, and autophagy is co-regulated with stemness, hedgehog, and immunomodulation. Our study demonstrated the presence of SRP and how, on the transcriptome level, TP53 and Wnt/β-catenin pathways are regulated to deliver the same cellular fate. Our study proves that SRP co-regulated with senescence-associated reprogramming (Wnt/β-catenin pathways) and TP53-p21 dysregulations originate from a common etiology and present a novel therapeutic target opportunity.

**Keywords:** keloids; senescence-resumed proliferation; Wnt/β-catenin pathways

## 1. Introduction

Senescence is most known for its replication cessation role, coined in the 1950s as "replicative senescence", where cell culture fails to expand under extended passages. Similar to replicative senescence that stems from aging, another term, "premature senescence", arises from oncogenic insults in which damaged cells prone to oncogenic activation activate the senescence program as a cellular protective measure. Premature senescence reinforces overall macro-integrity by forcing cells subjected to irreparable insults into dormancy through executing cell cycle arrest in the G0 phase. Nevertheless, the therapeutic effect of senescence is often limited in clinical reality. Therapy-induced senescence is often temporary before developing into a more aggressive tumor relapse. In the recent decade, researchers have begun to grasp that senescence is a dynamic and evolving phenomenon. Senescence may be installed and released/escaped at various time points and with various treatments. The novel concept, published multiple times in Nature, stresses that the senescence program may be spontaneously released/escaped. Damaged oncogenic cells that have undergone senescence fail to maintain their dormancy and resume their proliferative abilities. Milanovic et al. emphasized a novel research ground where dormant senescence cells are metabolically active, generating signals that enhance malignancy, therapy-resistance, and cancer stem cell (CSC) properties [1–5]. Cancer stem cells had increased glycolysis and glutamate and fatty acid (FA) catabolism, which fueled the tricarboxylic acid (TCA) cycle, allowing it to generate more ATP. Signaling pathways involved in metabolic reprogramming in cancer stem cells, such as the PI3K-AKT-mTOR pathway, were also significant [5]. Although it seems paradoxical that senescent cells that resume proliferation acquire enhanced oncogenic properties, Cruickshanks et al. demonstrated that methylation changes in senescence resemble cancer [6–8]. DNA methylation and associated destabilization of genome integrity are mediated by gains and losses of methylation in senescence that are qualitatively oncogenic-prone [6]. The altered methylation pattern

induced by senescence is retained in cells that are released from senescence. Despite resuming proliferation, previously senescent cells do not show methylation patterns that are reversed from their original proliferating patterns. The permanent epigenetic writings, methylome of senescent cells that are oncogenic, suggest that the senescence program is a double-edged sword and explains how, via epigenetic wiring, the senescent-released cells acquire a more aggressive phenotype.

The distinction between "irreversible senescence" and "senescence-like arrest" is pivotal in addressing the suitable timing of senolytic involvement in anticancer therapy [9]. Treatment-induced senescence (TIS) is acknowledged in tumors upon chemotherapy or radiotherapy challenge. Aside from the cytotoxic treatments, TIS can also be triggered by PTEN inhibitors, MDM2-p53 disruptors, epigenetic modulators, telomerase inhibitors, CDK inhibitors, AURK inhibitors, and PLK1 inhibitors. On the basis of senescence initiation, telomere erosion, hyperproliferative states, and reactive oxidative stress are well-established inducers. The succeeding molecular mechanisms, encompassing DNA double-strand break response, p53-p21 activation, p16-RB activation, autophagy, and NF-κB signaling, will further enhance cell cycle exit [10]. Of note, senescence-associated secretory phenotype (SASP), encompassing several pro-inflammatory cytokines, chemokines, growth factors, and matrix remodeling enzymes, will be generated from the aforementioned mechanisms. Some SASPs, such as IL6 and IL8, can shape the microenvironment via autocrine or paracrine mechanisms to elicit cell-autonomous or non-cell-autonomous senescent events [11]. Furthermore, it is context-dependent that the SASPs are beneficial or detrimental to the host immune network in TIS. CSF1, CCL2, CXCL1, IL-15, and IL-6 represent immunosupportive SASPs, which are capable of recruiting macrophages, neutrophils, and NK cells and polarizing pro-inflammatory M1 macrophages, collectively leading to the elimination of cancer cells [12]. On the contrary, senescent cancer cells (SnCs) can build an immunosuppressive niche through myeloid-derived suppressor cell (MDSC)-soliciting SASPs, such as CXCL1, CXCL2, GM-CSF, and M-CSF. From the perspective of immunogenic roles rendered by SASPs in TIS, escape from the TIS state can be a route of cancer immune editing. Acquisition of stemness is the center of the molecular basis of cancer senescence exit. Milanovic et al. blame canonical Wnt/β-catenin signaling (nuclear β-catenin activation), activated in therapy-induced senescence, as the essential driver of the enhanced tumor initiation capacity exhibited by senescence-resumed proliferation (SRP) tumor cells [1]. The β-catenin expression is the hallmark of Wnt/β-catenin signaling proposed to play a role in cancer stem cell properties, increased tumor resistance to therapy, and tumor malignancy [13,14]. Wnt/β-catenin is implicated in the physiologic machinery generating embryonic development, polarization, and also pathological events of CSC [14]. Milanovic et al. showed that multiple tumor lineages undergoing senescence acquired permissive (H3k9me3) and repressive (H3K27me3) histone chromatin remodeling that increased permissive transcription Wnt-stemness-related gene expression and the decreased repressive epigenetic control of these genes [1]. Histone marks have been proposed to be permanent despite tumors re-gaining cell proliferation. Apart from the epigenome, authors found that Wnt/β-catenin signaling and hTERT on the proteomic level are co-regulated in a positive feedback loop, yielding enhanced telomerase activity, CSC-like traits, and therapy resistance [15]. To combat the unwillingly occurring senescent cancer cells (SnCs), the "one-two punch therapy" is proposed as a sequential treatment with a senescence-promoting drug first, followed by senolytic agents that specifically eliminate SnCs [16]. Senolytic agents take advantage of the upregulated anti-apoptotic effectors in TIS cancer cells. Potential drugs under exploration include Navitoclax, ARV-825, AZD8055, ABT-737, and a cocktail combination of dasatinib and quercetin (D+Q). Navitoclax (ABT-263) is a BCL-2 inhibitor that manages the elimination of doxorubicin or etoposide-induced SnCs via dampening the BCL-$X_L$–BAX interaction [17,18]. Given that SASP content diversity and SnC plasticity are variable and context-dependent, senolytic effects of navitoclax have also been examined under plentiful clinical scenarios of TIS. For example, malignant meningioma cells entered cellular senescence (proven by the detection of increased SA-β-gal activity) upon conjunctive therapy

of gemcitabine and irradiation. Accordingly, navitoclax can boost the anti-tumor effect by halting the anti-apoptotic activity of these TIS meningioma cells [19]. Mouse oral squamous cell carcinoma gains senescent traits after cisplatin treatment. Administration of navitoclax can induce cell death on these TIS head and neck cancer cells, subsequently [20]. Additionally, in a non-chemotherapy scenario, prostate cancer (PCa) cells will develop cellular senescence when treated with androgen-deprivation therapies, including bicalutamide and enzalutamide. These TIS PCa tumor cells contribute to castration resistance. Navitoclax mediates the clearance of ADT-promoted senescent PCa cells and dampens the androgen-independent proliferation [21]. Dasatinib (D) is an Src/tyrosine kinase inhibitor, and quercetin (Q), is a natural flavonoid that binds to BCL-2 and modulates transcription factors, cell cycle proteins, pro- and anti-apoptotic proteins, growth factors, and protein kinases. The progression of several diseases has been demonstrated to be halted by the D+Q cocktail. For example, in idiopathic pulmonary fibrosis, physical function showed the most consistent improvement following D+Q. The 6-min walk distance (6MWD), 4-m usual gait speed, timed 5-repetition chair-stands, and short physical performance battery (SPPB) scores significantly improved one week after the drug was administered [22]. Moreover, quercetin itself can serve as an osteoporosis and atherosclerosis [23] protector. The effect of bone protection is present at the levels of elevated bone mineral density at variable sites, increased osteoid volume and surface, decreased erosion area, downregulated osteoclast population, and several functional aspects, including improved maximum power, energy, and load [24]. The senolytic cocktail D+Q has controversial pre-clinical results in the field of TIS cancer cell elimination. In a radiation therapy-resistant melanoma mouse model, D+Q could re-establish the susceptibility of melanoma cells by tuning down the cellular senescence marker SA-β-gal [25]. Nevertheless, doxorubicin-induced senescent hepatocellular carcinoma cells are not sensitive to D+Q treatment on the level of repressing SA-β-galactosidase expression [26].

Escaping from the cell cycle arrest and yielding tumor relapse, the evolved tumor's association with enhanced expansion ability may be a universal relationship that may be studied/predicted systematically. Previously, researchers have been limited in predicting the subsequent mutation paths as the evolutionary selection of tumor clones and DNA damage location is random. Thus, keloids serve as an advantageous and introductory disease model for studying the SRP phenomenon, as its disease etiology goes beyond the scope of more advanced uncontrollable and unmanipulable concepts of random genomic instability and clonal selection. Universal and predictable phenomena across tumor types suggest an identifiable target for therapeutic translation. Thus, to study such phenomena in a simplified tumor model, we used keloid, a benign proliferative scar that does not mutate or metastasize but may still relapse despite a clean tumor margin and presents with CSC-like properties and acquired therapy resistance [27].

Numerous SRP cellular findings are consistent with keloid findings. Current literature blamed the dysregulation of tumor suppressor protein TP53 [28–30], the increase of β-catenin [31–34], the embryonic stem cell expression [35], and the absence of senescence (TP53-p21 axis) maintenance for catalyzing keloid formation [36].

Nevertheless, authors fail to recognize that these diverse cellular mechanisms are merely products of common upstream signaling from senescence. Could it be possible that the reprogramming and stemness observed in this literature was bred from the same mechanism of SRP that we have delineated previously? It seemed that numerous phenotypes of keloid observed by different authors are not mutually exclusive and may all stem from cells acquiring a permanent reprogramming epigenetic mark during senescence. The current mainstream investigation of keloid is restricted to targeting downstream pathways and not the root of the pathogenesis. Numerous researchers targeting keloids delineate an observational outcome of phenotype without dissecting the true target. Our study aims to demonstrate that these observations stem from a single origin, that is, SRP.

No study has explored nulled role of senescence as the underlying pathophysiology keloid formation or integrated these manifestations as originating from a common mecha-

nism. Hence, SRP is a profoundly undermined research area that embodies unexplored and exciting novel pathophysiology of previously believed unassociated etiologies.

Therefore, we aim to dissect the role of senescence in keloids by investigating whether the dysregulated proliferative nature of keloids may be traced back to senescence. We anticipate that SRP fibroblasts will acquire a keloid phenotype characterized by an enhanced proliferation ability, increased migratory and invasive ability, and augmented tumor-initiating capacity. The results obtained will shift the dynamics of keloid investigation. Researchers will begin to approach keloid anti-senescence ability as not solely due to a unidirectional anti-apoptosis phenomenon but also due to alteration of its underlying cellular epigenome. Suppose our hypothesis that senescence acts as a common upstream signaling of diverse potential therapeutic targets is proven valid. In that case, one straightforward therapeutic value of not eliciting senescence may be that it yields a comprehensive tissue homeostasis restoration whereby multiple pathological pathways are concurringly interrupted.

## 2. Materials and Methods

RNA-seq Database: GEO (Gene Expression Omnibus), accession number: GSE44270, was used. GEO2R, an interactive online tool for identifying DEGs from GEO series, was used for pipeline analysis [37]. GEO2R can be used to differentiate DEGs between MC and AC subtypes. Probe sets in the absence of corresponding gene symbols were removed, as were genes with more than one probe set. Statistical significance for the dataset GSE103512 was established at $p \leq 0.05$ and log 2-fold change $\geq 1$. However, no fold change or p-value threshold was specified for the other datasets.

Hahn's dataset: For the purpose of isolating primary keratinocytes and fibroblasts, skin and scar tissues were acquired. In patients receiving scar excision surgery, keloid scars were removed, and in patients undergoing elective plastic surgery, normal skin samples were extracted. Keratinocyte and fibroblast primary cultures were made, and they were harvested for examination up to passage three. Three normal skin samples and nine keloid scars for nearby non-lesional keloid skin samples were collected and cultured. The quality of the RNA was checked using an Agilent 2100 Bioanalyzer after it had been isolated using RNeasy. The Vanderbilt Genome Sciences Resource at Vanderbilt University Medical Center carried out the labeling and hybridization to Affymetrix Human Gene 1.0 ST microarray chips. Database Source: A separate causative involvement in keloid disease is supported by the aberrant gene expression profile that keloid-derived keratinocytes display [38].

Cytoscape: Through Cyto-scape v.3.7.2, we conducted PPInetwork studies using the STRING database v.11. With the exception of the requirement that interactions be restricted to high-confidence ones, we used the multiple protein input option with all default values (0.700). In the network output, disconnected nodes were hidden, but not in the enrichment analysis. Based on their relationships with a greater number of other DEGs in the PPI network, the important genes were selected. Article Source: Cytoscape: a software environment for integrated models of biomolecular interaction networks. Shannon, Paul et al [39].

Plug-in Enrichment map: Article Source: "Enrichment Map: A Network-Based Method for Gene-Set Enrichment Visualization and Interpretation [40]."

Plug-in iRegulon:

Article Source: "iRegulon: From a Gene List to a Gene Regulatory Network Using Large Motif and Track Collections [41]."

GSEA: In this investigation, the gene set database h.all.v7.5.symbols.gmt (Hallmarks) was chosen. The functional differences of the gene between the keloid, keloid-prone fibroblast, and control groups were further explained using GSEA Each analysis involved 1000 different gene set arrangements. To categorize each phenotypic enrichment signaling pathway, the FDR q value, the normalized enrichment score (NES), and the nominal p value were developed. Article Source: "Gene set enrichment analysis: a knowledge-based approach for interpreting genome-wide expression profile [42]." Article Source: "Molecular signatures database (MSigDB) 3.0 [43]."

David bioinformatics resources: For functional enrichments based on Gene ontology

Article Source: "Systematic and integrative analysis of large gene lists using DAVID Bioinformatics Resources [44]."

Article Source: "Bioinformatics enrichment tools: paths toward the comprehensive functional analysis of large gene lists [45]."

RStudio: RStudio Team (2020). RStudio: Integrated Development for R. RStudio, PBC, Boston, MA URL http://www.rstudio.com/ (accessed on 15 February 2018).

## 3. Results

To establish that SRP phenomena observed in multiple cancer lineages are valid in the keloid disease model, we fully explored the keloid genomic landscape with RNA-seq data obtained from the GEO database. SRP is proposed to be a mechanism associated with the escape p21-mediated senescence and the activation of Wnt/β-catenin pathways that enhance malignancy. Thus, during the exploration, we purposely searched for senescence activation via the p53-p21 axis and Wnt/β-catenin signaling, including the TGF-β-HOX axis. Laying out the differential gene expression (DEG) comparing keloid versus control fibroblast demonstrates multiple HOX developmental genes and other novel target gene enrichment in keloid etiology, as shown in Figure 1A. Enriching the DEG from Figure 1A yielded hallmarks enrichment from gene ontology (GO) in Figure 1C. The enriched hallmarks encompassed numerous p53-related pathways, cell cycle, Wnt/β-catenin–TGF-β pathways, and immune modulations (Figure 1C).

Our search for reprogramming and stemness hallmarks revealed that Wnt-reprogramming hallmarks such as Wnt/β-catenin n signaling, hedgehog signaling, and TGF- β signaling enrichment confirm the importance of Wnt/β-catenin signaling in keloid formation (Figure 1C ). A heatmap of gene expression clustering among keloid groups, non-keloid fibroblasts from keloid-prone patients (susceptible), and control demonstrates nonspecific clustering of the control group in Figure 1B. The expression heatmap control group may mimic different groups' expression landscapes without eliciting pathogenesis and suggests that wide dysregulations are not associated with disease pathogenesis. Numerous dysregulations in keloid-prone individuals are insufficient in eliciting keloid pathogenesis in control patients. A smaller, key-determining switch, illustrated in Figure 1A, may be responsible for disease etiology. Although in keloid-prone individuals, keloid fibroblasts and non-keloid fibroblasts showed significant clustering of mutually exclusive expressions. This finding suggests that disease target elucidation may be most appropriately searched for by comparing keloid versus susceptible and highlights a potential future research direction. Thus, we acquire three pairs of case-control sequencing data, which are "non-lesion fibroblast in keloid patient versus normal fibroblast", "keloid fibroblast versus non-lesion fibroblast", and "keloid fibroblast in keloid patient versus normal fibroblast", respectively. DEG analysis and Venn diagram are achieved based on the GEO2R platform. Here, we lay out the three groups' differential gene expressions in Figure 2 and observe inclusive and exclusive expressions between groups.

Next, we aimed to lay out the enriched hallmark landscape using GSEA enrichment of the whole dataset and study their crosstalk and impacts (Figure 3). The landscape shows heavy intersections between TP53 and TGF-β signaling where the machinery to maintain cellular integrity through senescence, apoptosis, and autophagy are co-regulated with stemness, hedgehog, and immunomodulation. Intrigued by such a finding, we collapsed the landscape and recalculated the interactions (edges) most enriched in Figure 4a. Consistently, hallmarks such as TP53 regulation, hedgehog, and TGF-β-signaling remained as enriched node and edge. Interestingly, cyclin-dependent kinase inhibitor p21 degradation pathways are activated in conjunction with proteins that allow mitosis gate passage and may mediate the enhanced mitotic dysregulations in keloid, consistent with the SRP phenotype. The senescence marker p21 responsible for cell cycle arrest and senescence maintenance activates in conjunction with the subset of molecular pathways in hedgehog signaling, as

shown in Figure 4A, and points to senescence release (p21 degradation) co-regulation with stemness pathways.

**A**

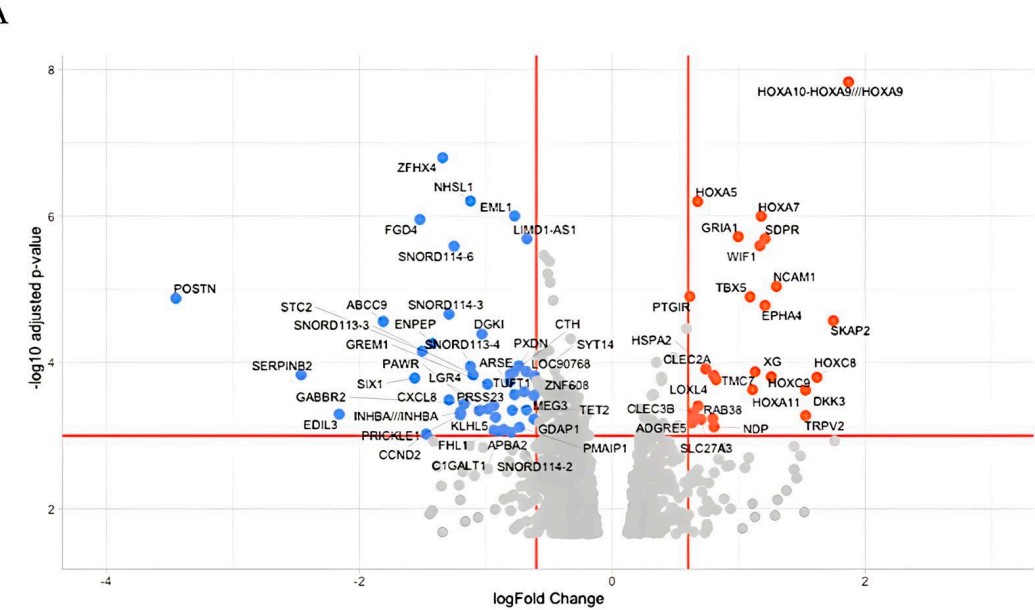

**B**

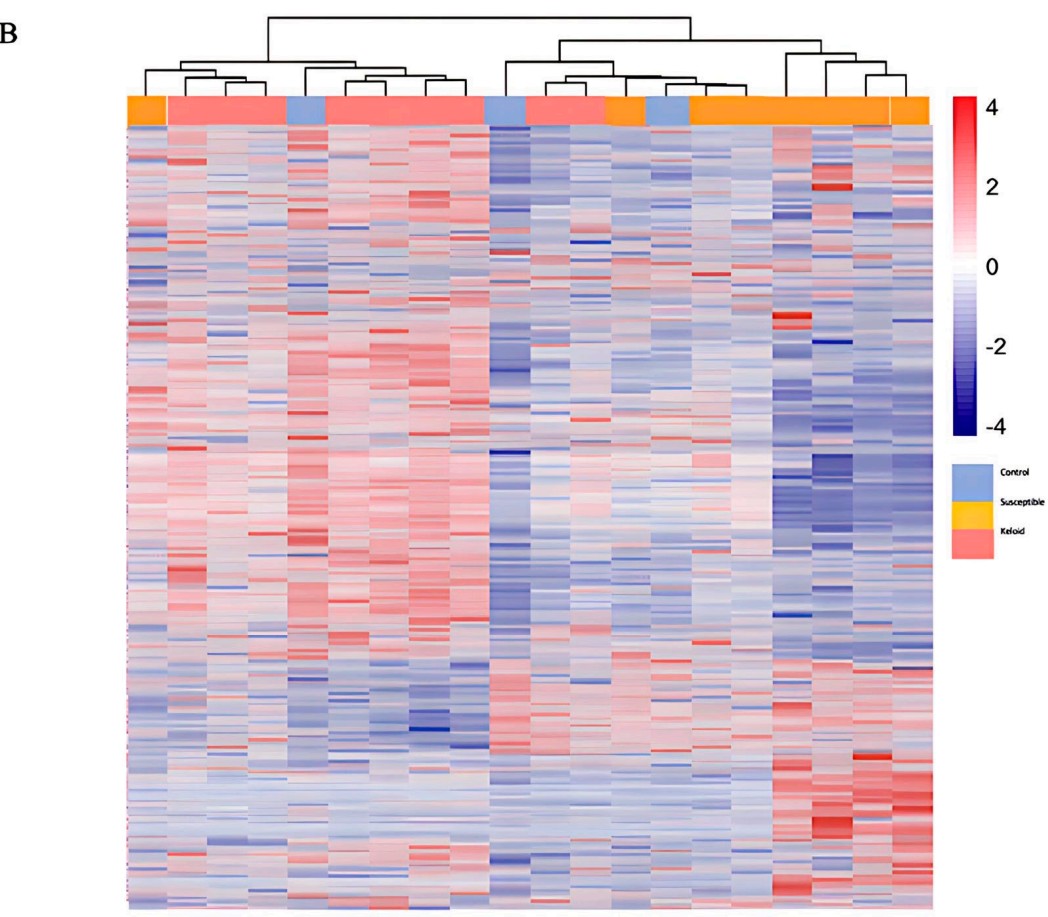

**Figure 1.** *Cont.*

**C**

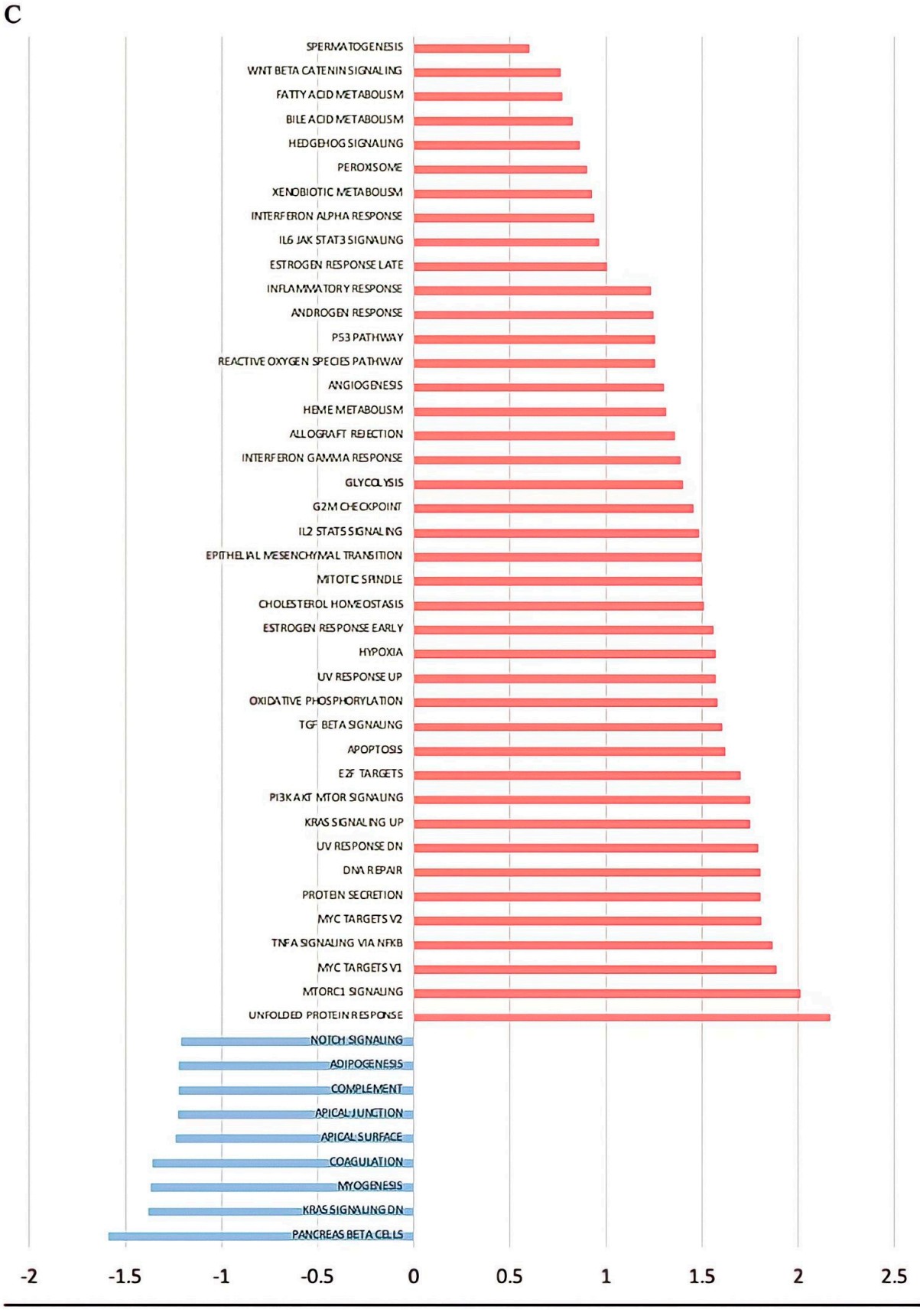

**Figure 1.** *Cont.*

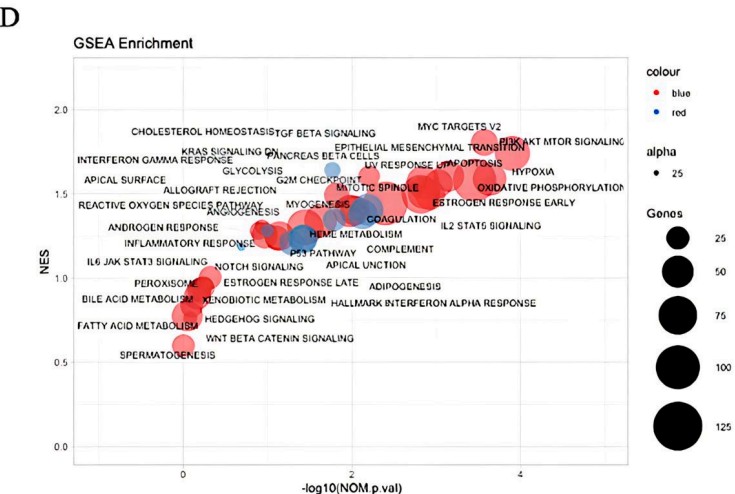

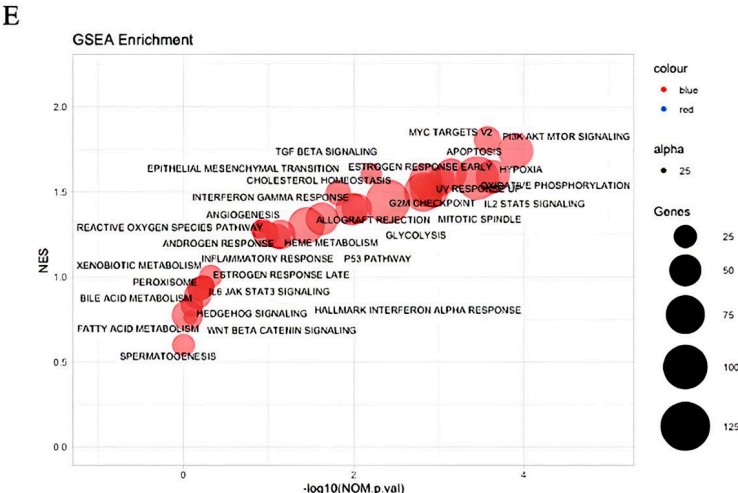

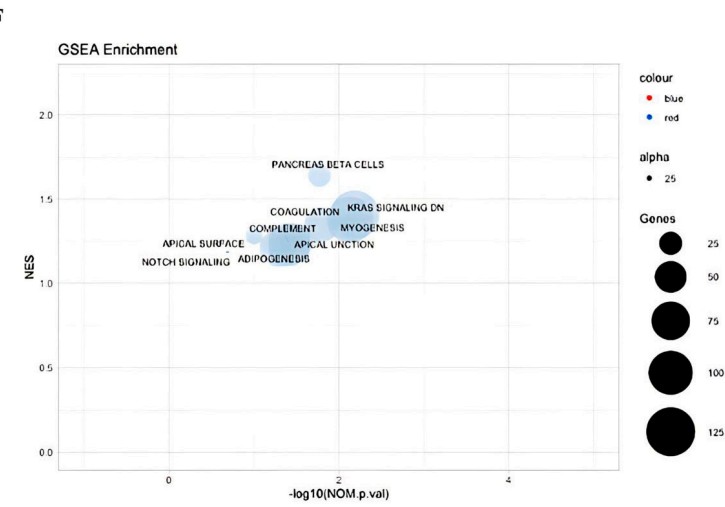

**Figure 1.** Genomic landscape of keloid. (**A**) Volcano plot of differential gene expression (DEG) of keloid versus control; (**B**) heatmap clustering of gene expression between control, susceptible, and keloid; (**C**) gene ontology hallmark enrichment of DEG; (**D**) overall bubble plot of gene ontology hallmarks; (**E**) increased hallmark enrichment in keloid; (**F**) increased hallmark enrichment in control, decreased enrichment in keloid.

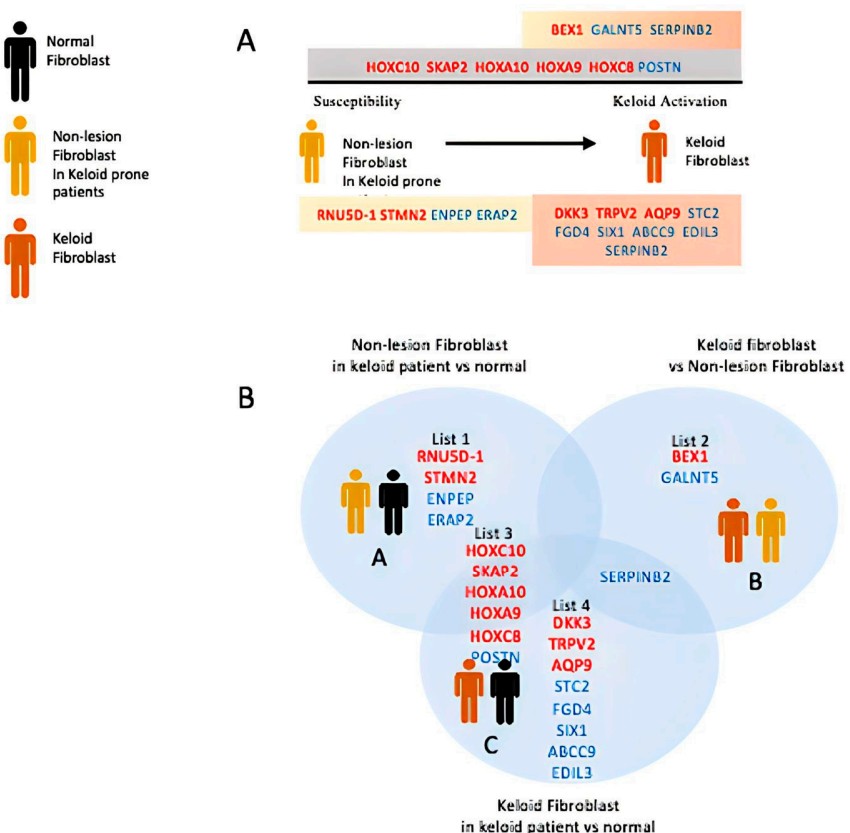

**Figure 2.** DEG of keloid, susceptible, and control. (**A**) The first row depicts exclusive gene expression of keloid/susceptible (list2); second gray row depicts the common underlying difference between keloid/susceptible with control (list3); and the bottom row (list1, 4) depicts the exclusive distinction of keloid/susceptible with control. (**B**) Venn diagram illustrating the above relationships.

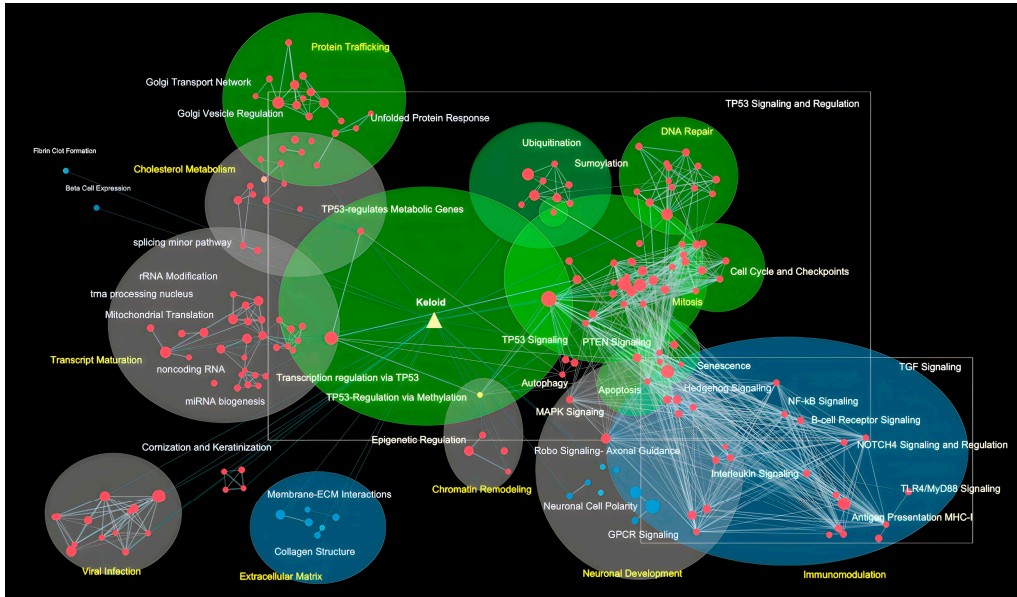

**Figure 3.** Gene ontology hallmark landscape of keloid. In the layout, we depicted the TP53-associated pathway (Green), Wnt/β-catenin signaling (blue), and miscellaneous (grey) with different colors and highlighted their complex interactions. Nodes depicted in red signify increased enrichment and those in blue signify decreased enrichment.

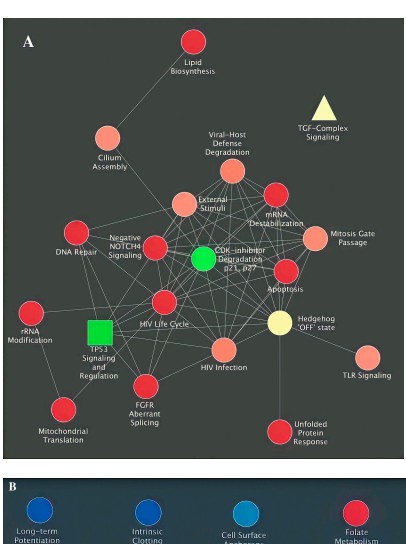

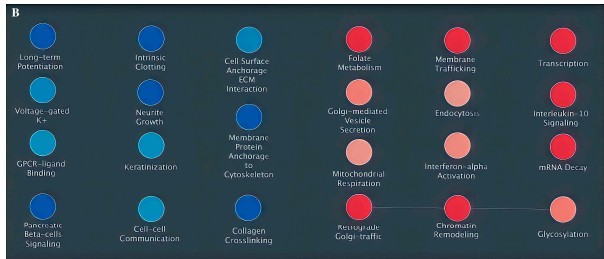

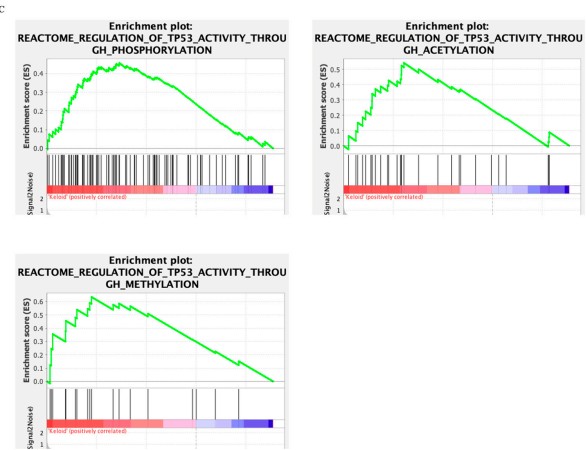

**Figure 4.** Collapsed clustering of GO-hallmark landscape. (**A**,**B**) Enriched interactions with TGF-β- signaling (yellow) and TP53, senescence (green); (**C**) GSEA enrichment of TP53 epigenetic modulations.

Nevertheless, the enrichment to TGF-β-signaling is dampened. It may suggest that their interaction is equally potent in deciding cell fate despite their interaction enrichment not reaching the significance threshold. In contrast, such findings may also indicate that their primary interaction is not mediated on the proteomic level but by other epigenetic and genomic methods and require different methodologies, such as ATAC-seq or CHIP-seq. Their associations may lie in the epigenetic framework and not in the genomic or proteomic study. Regardless, we investigated whether the TP53 activation may modulate epigenetic reprogramming by performing GSEA enrichment of TP53 epigenetic modulations in Figure 4B and showed that during keloid pathogenesis, TP53 modulated numerous epigenetic changes that may dictate subsequent phenotype change. A finding that is unsatisfactory to explain but hints that TP53 may modulate the permanent senescence epigenetic markers that resemble that of cancer.

Upon confirming that senescence, TP53, and TGF-β-signaling are of crucial importance on the landscape level (big-to-small filtering) to prevent bias, we then explored what these

narrowed-down hallmarks meant for the DEG expression (Figure 1A) level in Figure 5. We first allocated the DEG on the outer circle in increased expression levels (blue to red), calculated the enriched transcription factors of these DEG, and placed them in the inner circle in Figure 5A. Then, we aimed to see how numerous processes related to our study are highlighted in our DEG circle and found that the DEG is highlighted in our multiple target hallmarks in Figure 5B. This finding suggests that the proteins are heavily intertwined and orchestrated in the seemingly independent pathways. This finding proposes that the multiple previously studied keloid pathways may stem from one originating pathway/cause. To demonstrate again that the target hallmarks enriched in keloid etiology are not deliberately forced upon DEG, we show their GSEA enrichment in Figure 5C.

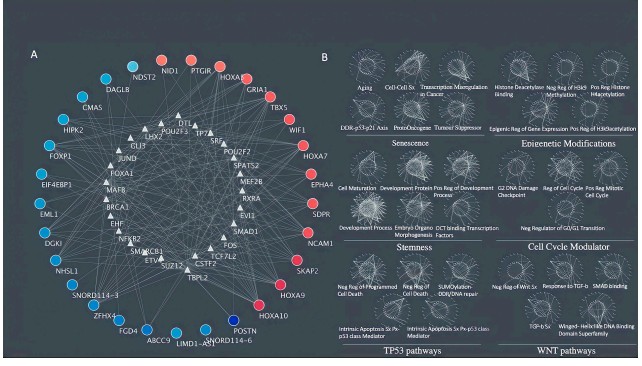

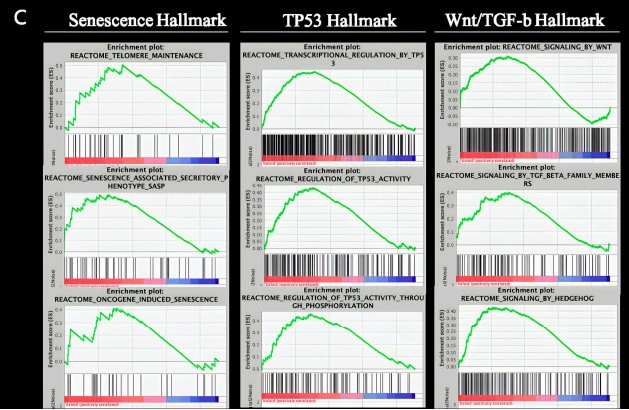

**Figure 5.** Target hallmarks are co-regulated in keloid etiology. (**A**) DEG of keloids versus control is displayed in outer circle and their enriched transcription factors in the inner circle; (**B**) The same DEG-TF interactions are present in these regulated target hallmark; (**C**) GSEA confirmation of target hallmark enrichment.

## 4. Discussion

According to the RNA-seq data in the GEO database, we obtained DEG comparing keloid versus control fibroblasts and revealed several key genes, such as HOXA10 and HOXA9, that enriched during the genesis of keloid. TGF-β signaling, Wnt-β-catenin signaling, p53 pathway, hedgehog signaling, and several endogenous stress response reactions are the prominently enriched hallmarks of gene ontology (GO).

Endogenous stress is emphasized by hallmark enrichments such as inflammatory response, ROS, metabolism reprogramming, hypoxia, UV response, and oxidative phosphorylation hallmarks that ought to activate the senescence program [46]. To counteract these enrichments, hallmarks such as unfolded protein response, autophagy, DNA repair, protein secretion signaling, and apoptosis are activated.

Subsequently, we demonstrated the GO hallmark landscape of keloid, which presented an intensive overlap between the TP53 and TGF-β networks. These conjunctions encompass senescence, apoptosis, autophagy, stemness, and hedgehog signaling, which prompted us

to recalculate the most enriched edges from a collapsed hallmark landscape. As a result, p21 degradation pathways and mitosis gate passage events are conjunctly activated, implying the existence of a compatible SRP feature of dysregulated mitosis in keloid biology.

We reviewed recent literature to elucidate how TP53 may modulate TGF-β/Wnt-β-catenin signaling. Apart from epigenetic modulations of histone binding on Wnt clusters [1], we found that TP53 and HOX, both serving as transcription factors, mutually co-influence the transcriptome.

HOX is a potent transcriptional activator of p53 [47,48], and consensus HOX binding sites are present in the p53 gene-promoter region [49]. Genome-wide profiling of decoding proteins transcribed by TP53 include the HOX gene family where p53 binding sites coincide with stem cell transcription factors, such as OCT4, NANOG, and H3K27me3 pluripotency [50].

Future research should consider demonstrating the senescence features in the suitable patient cohort. The p21 expression pattern in keloid samples can represent the sustaining and release of senescence. Intriguingly, p21 can form an effector network named p21-activated secretory phenotype (PASP) that determines the dynamics of cell fate. Long-term p21 induction results in the establishment of immunosuppressive traits of mouse embryonic fibroblasts (MEFs) [51]. M1 macrophage polarization is supported by CXCL14, one of the PASPs, and p21 is also known as a senescence-apoptosis switch in non-small cell lung cancer (NSCLC) [52]. This switch is regulated by the ATM/miR-34a-5p axis.

Of note, the intrinsic discrepancies of the keloid tissue should be considered. Intralesional heterogeneity distinguish central and peripheral keloid regions on the levels of clinical manifestations, histological markers, and molecular events [53]. The periphery presents a more pigmented and erythematous lesion with an elevated height and stiffer content. The peripheral epidermal tissue has higher epithelial-to-mesenchymal transition potential as compared to the central part, as demonstrated by the elevated vimentin level. Additionally, higher cellularity and more apoptotic cells are found in the peripheral dermis. As for the peripheral fibroblast, it is known to have an increased proliferation rate and elevated p53, bcl-2, and MMP-1/2/3/9. Since the central and peripheral keloids have been proposed to be heterogeneous, we propose that SRP phenomena are more pronounced in the peripheral keloid than in the centre keloid. Intralesional keloid heterogeneity shows that a raised peripheral margin actively invades the surrounding skin, while the depressed centre undergoes clinical regression [53,54]. The peripheral–central distinction is intensely explored where peripheral keloid is associated with hypercellularity [30,55–57], vascularity [58,59], and increased cellular activity [60], while the central keloid demonstrated hypocellularity, reduced vascularity, increased apoptosis, senescence [36], and inactivity.

Furthermore, an allocation of keloid samples according to the skin layer of the keratinocytes (reticular dermal fibroblast, papillary dermal fibroblast, and stratum basale) may aid in the minimization of the intra-sample comparison bias. On the other hand, as a hallmark of senescence is the acquisition of senescence-associated secretory phenotype (SASP), senescence and oncogenic potentials are propagated to neighboring cells. Literature has iterated the keratinocyte–fibroblast crosstalk role in repetitively mediating keloid pathogenesis [61–63]. Interestingly, in our study of fibroblasts, hallmark enrichment of increased cell–cell communication, SASP, golgi secretion hallmarks coupled with reduced keratinization, and cornification hallmarks were presented, which hints at the potential role of keratinocyte crosstalk. To confirm the idea of intercellular crosstalk underlying keloid biology, further investigation can utilize the p21 staining to differentiate the expression patterns between papillary dermis fibroblast and stratum basale keratinocytes.

## 5. Conclusions

Previous research has aimed to reinstate senescence in cells that are undergoing uncontrolled proliferation. Nevertheless, reinstating senescence is futile in disease pathogenesis that has only surfaced in recent years, where senescence is established to be dynamic and may be escaped to resume proliferation. Thus, pro-senescence therapeutics aiming to achieve senescence are limited. Furthermore, research has found that senescence is

not previously assumed to be dormant, but undergoing robust metabolic expressions and secretions that are oncogenic may lead to more aggressive relapse and neighboring tumorigenesis. Our study demonstrated the presence of SRP and how, on the transcriptome level, senescence-associated reprogramming (Wnt/β-catenin pathways) and TP53-p21 dysregulations originate from a common etiology. In short, this evidence stokes expectations of treating senescence-associated reprogramming as a future pharmaceutical target of keloid.

**Author Contributions:** Conceptualization, T.-Y.L. and C.-Y.W.; methodology, T.-Y.L.; software, T.-Y.L.; validation, C.-Y.W., C.-W.W., and T.-Y.L.; formal analysis, T.-Y.L.; investigation, C.-W.W.; resources, T.-Y.L.; data curation, T.-Y.L.; writing—original draft preparation, T.-Y.L.; writing—review and editing, C.-Y.W. and T.-Y.L.; visualization, C.-Y.W., and C.-W.W.; supervision, T.-Y.L.; project administration, T.-Y.L.; funding acquisition, T.-Y.L. All authors have read and agreed to the published version of the manuscript.

**Funding:** This research was funded by Academia Sinica (AS-TM-110-02-02).

**Institutional Review Board Statement:** Not applicable.

**Informed Consent Statement:** Not applicable.

**Data Availability Statement:** Not applicable.

**Acknowledgments:** The authors are grateful to Academia Sinica for the support of this investigation.

**Conflicts of Interest:** The authors declare no conflict of interest.

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
