# Peer review of "Role of Senescence-Resumed Proliferation in Keloid Pathogenesis"

_futurepharmacol, doi:10.3390/futurepharmacol3010014_

Round 1

Reviewer 1 Report

In this manuscript, the authors investigated that the presence of Senescence-resumed proliferation(SRP) and how, on the transcriptome level, senescence-associated reprogramming (Wnt/ β-catenin pathways) and TP53-p21 dysregulations originate from a common etiology. It’s interesting. At present, there are few researches on this aspect, so it has certain innovation. The manuscript is well-organized, however, some concerns need to be addressed.

1)     In Figure 1, some pictures are not clear, the author needs to revise.

2)     In Figure 2, the authors screened for differential gene expression. How did the author screen?

3)     In abstract and conclusions, the author didn’t clearly state the core conclusion of this article, so I strongly advised the author to revise it.

Author Response

Point 1: In Figure 1, some pictures are not clear, the author needs to revise.

Response 1: We will renew the size and resolution of figure1 to make it more readable. Thank you.

Point 2: In Figure 2, the authors screened for differential gene expression. How did the author screen?

Response 2 : The Venn diagram (Figure 2B) summarized the idea of DEG acquisition. At the beginning of analysis, we prepared three pairs of case-control sequencing data from the published database, which were “non-lesion fibroblast in keloid patient versus normal fibroblast,” “keloid fibroblast versus non-lesion fibroblast,” “keloid fibroblast in keloid patient versus normal fibroblast,” respectively. We then utilized the GEO2R platform to screened the differential expressed genes in each aforementioned case-control pair. Eventually, the output of Venn diagram was formed by the visualization of the overlap in significant genes (DEGs) among the three selected contrasts.

Point 3: In abstract and conclusions, the author didn’t clearly state the core conclusion of this article, so I strongly advised the author to revise it.

Dear editor, we've added a short paragraph in the conclusion section to state our core conclusion. Thank you.

Reviewer 2 Report

The manuscript entitled, “Role of Senescence-resumed Proliferation in Keloid Pathogenesis” is interesting. Overall, the data demonstrated that Senescence-resumed proliferation (SRP) is co-regulated with Wnt/β-catenin pathways and dysregulations in TP53-p21 signaling originating from a common etiology, and could be explored as a novel therapeutic target for keloid pathogenesis. Overall, the studies are nicely designed and presented. I just have a minor suggestion. Please keep the sentence, “Article Source: Bioinformatics enrichment tools: paths toward the comprehensive functional analysis of large gene lists.”, together.  

Author Response

Point1: Please keep the sentence, “Article Source: Bioinformatics enrichment tools: paths toward the comprehensive functional analysis of large gene lists.”, together.  

Response1: We've revised the paragraph as your advice, thank you!